# Mitochondria delay action potential propagation
Ann M. Castelfranco ⬤ [1] & Pepe Alcami ⬤ [2,3] ✉

The internal resistance of axons to ionic current flow determines action potential conduction velocity. Although mitochondria support axonal function, axons have been modeled as organelle-free cables, and mitochondrial impact on conduction velocity, specifically by increasing internal resistance, remains understudied. We combine computational modeling and electron microscopy of forebrain premotor axons controlling birdsong production. Modeling shows that when the propagating action potential in an unmyelinated axon encounters a mitochondrion, conduction velocity decreases, delaying the action potential by tenths of microseconds to microseconds, an effect that is stronger in small axons. Axonal mitochondria thereby induce conduction inhomogeneities, accumulating total delays of tenths of milliseconds to ~ a millisecond over 3 millimeters-long axons, in the range of the temporal precision of these neurons. Thus, by partially occupying the axoplasm, mitochondria constrain information processing in vertebrate small-diameter axons. Our model should permit future investigations on the impact of mitochondrial axonal plasticity on action potential temporal coding.

The fine-tuning of axonal biophysical properties ensures the timely propagation of action potentials necessary for the proper functioning of neural circuits[1–5]. A fundamental property of action potential propagation, conduction velocity, depends on the morphological and biophysical properties of axons and, when myelinated, on myelin. Axonal morphology constrains the passive properties of axons, a phenomenon that is currently well understood[4,6]. However, a property of biological cables, and its impact on conduction velocity, has remained undetermined: the presence of intracellular organelles, in particular, mitochondria.

In comparison to non-biological cables, biological cables such as axons require the supply of energy all along the cable to maintain their function[7–9]. Mitochondria provide axons with energy, required to maintain resting potentials and propagate action potentials, as well as for housekeeping and synaptic function. By occupying intracellular volume, mitochondria are expected to reduce cytosolic cross-sectional area and increase the axial resistance to current flow, thereby decreasing the conduction velocity of propagating action potentials[1,10]. However, their impact on action potential propagation remains poorly studied, and axons have so far been modeled as cables without organelles[11].

In this article, we investigated the impact of mitochondria on action potential conduction velocity in the characteristic thin, unmyelinated axons found in vertebrate brains, with diameters in the range of tenths of micrometers[8,12–14]. For instance, Braitenberg[12] measured an average axonal diameter below 0.3 μm in the mammalian neocortex, and Aboitiz et al.[14]

found corpus callosum unmyelinated axon diameters to be in the range of 0.1 to 1 μm. We focused on the premotor pathway involved in the fast control of vocal muscles that produce birdsong, that is, the axons belonging to the principal cells of the song nucleus HVC (formerly known as 'high vocal center', used here as a proper name) that project to the RA (robust nucleus of the arcopallium), the $HVC_{RA}$ cells. $HVC_{RA}$ cells fire action potentials with submillisecond precision in high-frequency bursts during singing[15]. Thus, this pathway provides a good system in which to investigate delays in action potential propagation that could be induced by mitochondria.

## Results

### Mitochondrial occupancy of unmyelinated axons depends on axonal diameter

We examined, using transmission electron microscopy, the pathway connecting two song nuclei involved in birdsong production, HVC and RA in the canary (*Serinus canaria*) (Fig. 1A). This motor pathway is formed by a mixture of myelinated and unmyelinated axons, spatially clustered together in axon bundles (Fig. 1B).

We measured the fraction of the axonal cross-sectional area occupied by a mitochondrion, which we termed the "cross-sectional mitochondrial occupancy" of the axon, by dividing the area of a mitochondrion by the area of its axon, for unmyelinated axons in mitochondrion-containing axonal radial sections. Representative examples of axons of different sizes are shown in Fig. 1C. The cross-sectional mitochondrial occupancy ranged

[1]Békésy Laboratory of Neurobiology, Pacific Biosciences Research Center, University of Hawaiʻi at Mānoa, Honolulu, HI, USA. [2]Division of Neurobiology, Faculty of Biology, Ludwig-Maximilians-Universität München, Planegg - Martinsried, Germany. [3]Department of Behavioural Neurobiology, Max Planck Institute for Biological Intelligence, Seewiesen, Germany. ✉e-mail: alcami@bio.lmu.de

**Article**

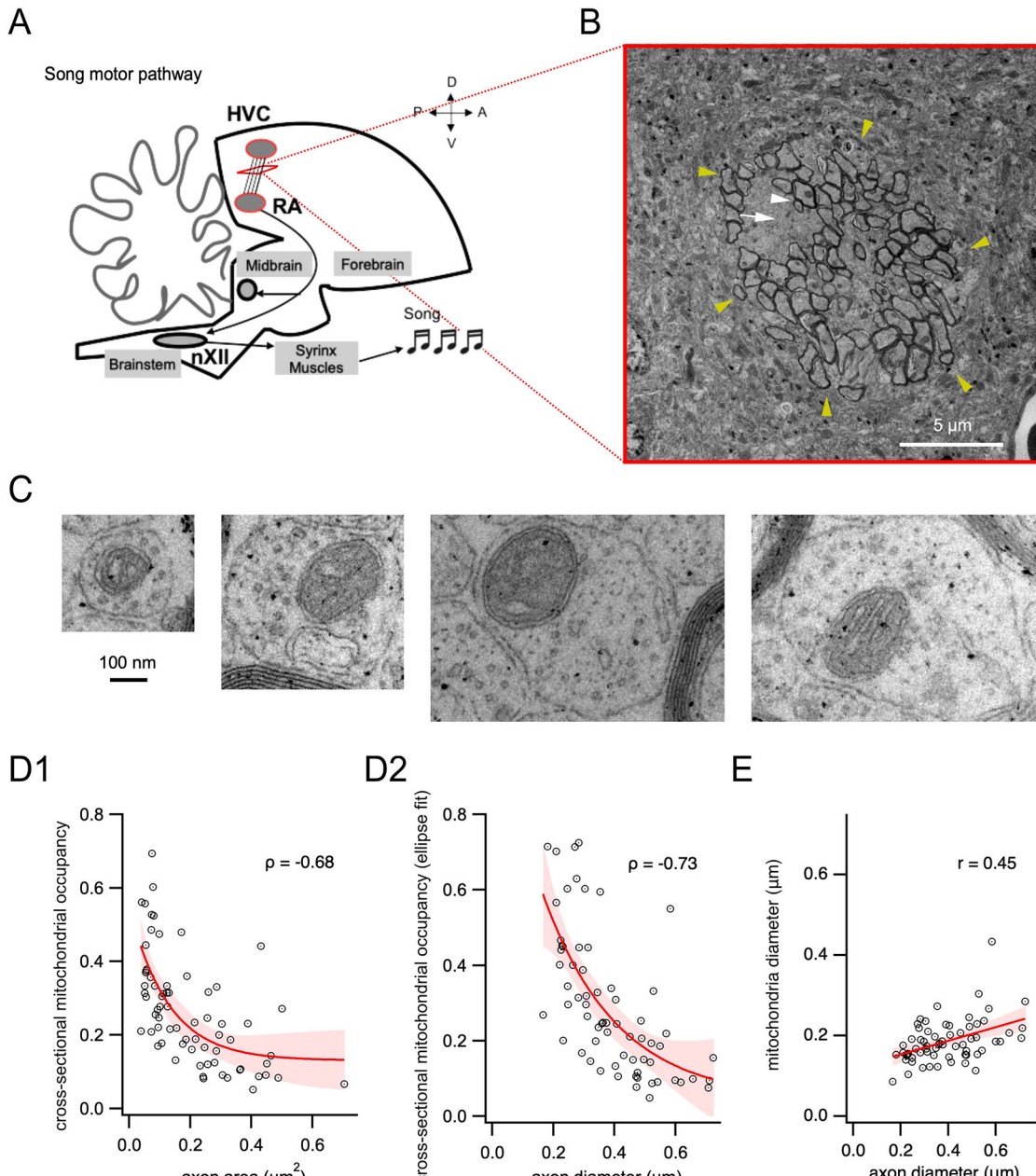

**Fig. 1 | Mitochondrial occupancy of unmyelinated axonal cross-sections between nuclei HVC and RA decreases with axon diameter. A** Schematic representation of the axons running between HVC and RA in the song motor pathway. The song motor pathway originates in HVC within the nidopallium, continues to the downstream nucleus RA in the arcopallium, whose neurons project onto nucleus nXII in the brainstem, which in turn projects to the vocal muscles that form the songbird vocal organ, the syrinx. Modified from[50]. **B** Example of an axon bundle (yellow arrows) formed by axons from the pathway between HVC and RA, imaged with transmission electron microscopy. Axons comprise myelinated axons (white arrowhead) and unmyelinated axons (white arrow). **C** Representative examples of axonal cross-sections of different sizes containing mitochondria. **D1** Occupancy of axonal cross-sections by mitochondria, measured as the ratio of mitochondrial to axonal areas, plotted as a function of measured axon area ($n = 69$). An exponential fit (±95% C.I.) of the data is shown in red. Spearman correlation coefficient ρ is reported in the figure. **D2** Occupancy of axonal cross-sections by mitochondria, measured as the ratio of the area estimated from a disk whose diameter was set to the minor axis of an ellipse fitted to the mitochondrion to the area estimated analogously for the corresponding axon, as a function of axon diameter ($n = 66$). Spearman correlation coefficient ρ is reported in the figure. An exponential fit (±95% C.I.) of the data is shown in red. **E.** Mitochondrion diameter plotted as a function of axon diameter ($n = 66$). A linear fit (±95% C.I.) of the data is plotted in red. Pearson correlation coefficient r is indicated.

from 0.052 to 0.693 (Fig. 1D1), with an average of 0.264 ± 0.147 ($n = 69$ cross-sections of axons containing mitochondria from two canaries; mean ± standard deviation). Alternatively, we fitted ellipses to mitochondria and axons and calculated areas based on the minor axis of the ellipse, assuming the axons were cylindrical, correcting for planes not perfectly orthogonal to axon bundles[13]. This quantification gave similar results, with cross-sectional mitochondrial occupancies ranging from 0.048 to 0.725 and averaging 0.293 ± 0.186 (Fig. 1D2).

Mitochondria-containing unmyelinated axon diameters ranged from 0.166 μm to 0.724 μm ($n = 69$), averaging 0.396 ± 0.139 μm. In order to examine whether cross-sectional mitochondrial occupancy varied depending on axon diameter, we plotted cross-sectional mitochondrial occupancy as a function of axon diameter (Fig. 1D2). Interestingly, cross-sectional mitochondrial occupancy, both calculated from area measurements on electron micrographs and based on ellipse fits, showed a negative correlation with axon area and diameter (Fig. 1D1, D2, Spearman rank

correlation test, $P = 1.7 \times 10^{-10}$, $\rho = -0.68$; and $P = 2.8 \times 10^{-12}$, $\rho = -0.73$ respectively). That is, a larger fraction of the axon cross-section was covered in smaller axons relative to larger axons. Whereas on average mitochondrial occupancy accounted for 46.2 ± 17.1% of the axon cross-section (by the fitted ellipse method) for axons up to 0.3 μm in diameter, this fraction decreased to 21.4 ± 13.3% for axons larger than 0.3 μm in diameter (n = 21 and 45 respectively, Mann-Whitney test, $U = 106$, $P = 4.3 \times 10^{-8}$).

These results suggested little or no scaling of mitochondria with axon size. Indeed, we confirmed that the distributions of mitochondria and axon diameters showed a small correlation (Fig. 1E), which could be fitted by a linear function with slope 0.17 ($P = 0.00018$, $r = 0.45$).

### Model for mitochondrion-containing axonal sections

To model the impact of a mitochondrion on conduction velocity, we made the assumption that the mitochondrion acts like a region of high resistance to axial current flow leaving unchanged the axolemma resistance and the axonal cross-section, and thereby, the axonal membrane capacitance. Consider a section of an axon that contains a cylindrical mitochondrion (Fig. 2A, B1, B2). Since the mitochondrion doesn't completely fill the cross-section of the axon, there are two current paths through the core of the cylindrical axon: one passes through the cytoplasm, avoiding the mitochondrion, and the other passes through the mitochondrion (Fig. 2C1). This can be described by a circuit with two impedances in parallel, which can be approximated by two resistors in parallel, where $r_{ax}$ (Ω) is the resistance of the path through the cytoplasm and $r_{mit}$ (Ω) is the resistance of the path through the mitochondrion[16] (Fig. 2C2). These resistances can be replaced by an equivalent resistance $r_{eq}$ given by:

$$r_{eq} = r_{ax}r_{mit}/(r_{ax} + r_{mit}) \qquad (1)$$

Rewriting the equivalent resistance, $r_{eq}$, in terms of the intracellular resistivity, $R_{eq}$ (Ω cm), gives:

$$r_{eq} = R_{eq}L/A \qquad (2)$$

where $A$ is the cross-sectional area of the cylindrical axon and $L$ is the length of the section containing the mitochondrion, which we define to be the length of the mitochondrion.

The cross-sectional area, $A$, can be partitioned into the area taken up by the mitochondrion, $A_{mit}$, and the area free from the mitochondrion, $A_{ax}$.

Let $p$ be the proportion of $A$ that is taken up by the mitochondrion, then

$$A_{mit} = pA \text{ and } A_{ax} = (1 - p)A \qquad (3)$$

Combining Eqs. 2 and 3 gives:

$$r_{ax} = R_{ax}L/(1 - p)A \qquad (4)$$

where $R_{ax}$ is the cytoplasmic resistivity of the axon, and similarly,

$$r_{mit} = R_{mit}L/pA \qquad (5)$$

Hence, combining Eqs. 1, 4 and 5, the equivalent resistivity of the section is:

$$R_{eq} = R_{ax}R_{mit}/(pR_{ax} + (1 - p)R_{mit}) \qquad (6)$$

Thus, given estimates for the intracellular resistivity of the axon and the resistivity of a mitochondrion, we can estimate the combined resistivity in terms of the cross-sectional mitochondrial occupancy ($p$). In particular, if the resistivity of the mitochondrion is sufficiently large that the current path through the mitochondrion can be ignored, then $R_{eq} = R_{ax}/(1-p)$, the limit of $R_{eq}$ as $R_{mit}$ goes to infinity. So, for a given value of $p < 1$, $R_{eq}$ saturates as $R_{mit}$

gets large. On the other hand, if the mitochondrion fills the axonal cross-section, then $p = 1$ and $R_{eq} = R_{mit}$.

In order to apply this model to $HVC_{RA}$ axons, we need an estimate for the resistivity of a mitochondrion. This is not a particularly well-defined quantity, but we can make a rough estimate for a model cylindrical mitochondrion 1 μm in length with a typical diameter of 0.2 μm (mitochondria diameters are illustrated in Fig. 1E). If we assume that the resistance across a unit area of mitochondrial outer membrane is similar to that of the axolemma, on the order of $10^4$ Ωcm², then for a cylindrical mitochondrion 0.2 μm in diameter the membrane resistance per unit length is $\sim 1.6 \times 10^8$ Ωcm[17]. So the resistance of a 1 μm long mitochondrion is $\sim 1.6 \times 10^{12}$ Ω. Thus, the resistivity of the model mitochondrion is ~5 MΩcm. This is a crude estimate that ignores much of the structure of the mitochondrion. However, Jonas et al.[18] using a patch clamp technique to record from mitochondria in the presynaptic terminal of the squid found only a small channel conductance of ~28 pS in the quiescent terminal. This corresponds to a membrane resistance of more than $\sim 4 \times 10^{10}$ Ω for the patch. In order to use this resistance to estimate the resistivity, we need the dimensions of the resistor. The diameter of the tip of the patch electrode was ~0.2 μm, but whether the recording was from the surface of the mitochondrion or extended to inner membranes was unclear. For a mitochondrion of 1 μm length, this gives an estimated resistivity of at least $10^5$ Ωcm. In the simulations, we used the conservative estimate of $10^4$ Ωcm for the resistivity of a mitochondrion and 100 Ωcm for the intracellular resistivity of the axon. However, a larger value for the mitochondrial resistivity would have only a small effect on the results, since for a given occupancy ($p$), the equivalent resistivity ($R_{eq}$) saturates for large values of mitochondrial resistivity. For example, for $p = 0.4$, $R_{eq} = 165.6$ Ωcm for mitochondrial resistivity = $10^4$ Ωcm and intracellular resistivity = 100 Ωcm, and $R_{eq} = 166.6$ Ωcm when mitochondrial resistivity is increased to $10^5$ Ωcm.

### Mitochondria induce a local decrease in action potential conduction velocity

We used the computational model to investigate the impact of the cross-sectional mitochondrial occupancy on action potential propagation in unmyelinated axons. Since conduction velocity[10] and cross-sectional mitochondrial occupancy (Fig. 1) depend on axon diameter, we simulated action potential propagation for different axon diameters. We first explored the local effect of a single mitochondrion on conduction velocity. In order to measure mitochondrial length on the longitudinal axis, we prepared longitudinal sections from axon bundles (Fig. 2A, B1). The longitudinal length of a mitochondrion ranged from 0.242 to 1.882 μm measured as the major axis from an ellipse (average: 0.622 ± 0.281 μm, n = 55, Fig. 2B2). Thus, we report the local impact of a mitochondrion on action potential conduction for a typical mitochondrion length of 0.6 μm (Fig. 2B2).

We compared the conduction velocity of an action potential propagating along the membrane of a 0.6 μm long cylindrical axon containing a mitochondrion (Fig. 2C1, D) with that of the same model axon devoid of mitochondria (i.e., cross-sectional mitochondrial occupancy = 0). Local conduction velocity decreased with increasing mitochondrial occupancy and decreasing axon diameter. For different typical cross-sectional occupancies and axon diameters, conduction velocity decreased by ~0.11 m/s (36%) for small 0.2 μm-diameter axons with 0.6 cross-sectional occupancy, by ~0.06 m/s (13%) for medium 0.4 μm-diameter axons with 0.25 cross-sectional occupancy and by ~0.04 m/s (8%) for larger 0.6 μm-diameter axons with 0.15 cross-sectional occupancy. Thus, smaller axons are more susceptible to a decrease in conduction velocity induced by mitochondria.

We then computed the additional propagation delay through the mitochondrion-containing section for different axon diameters (Fig. 2E). Conduction delay increased with increasing mitochondrial occupancy and decreasing axon diameter. For typical cross-sectional mitochondrial occupancies in the measured biological range (Fig. 1), the additional delay caused by a single mitochondrion was ~1.1 μs for small 0.2 μm-diameter axons with 0.6 cross-sectional occupancy, ~0.2 μs for medium 0.4 μm-diameter axons

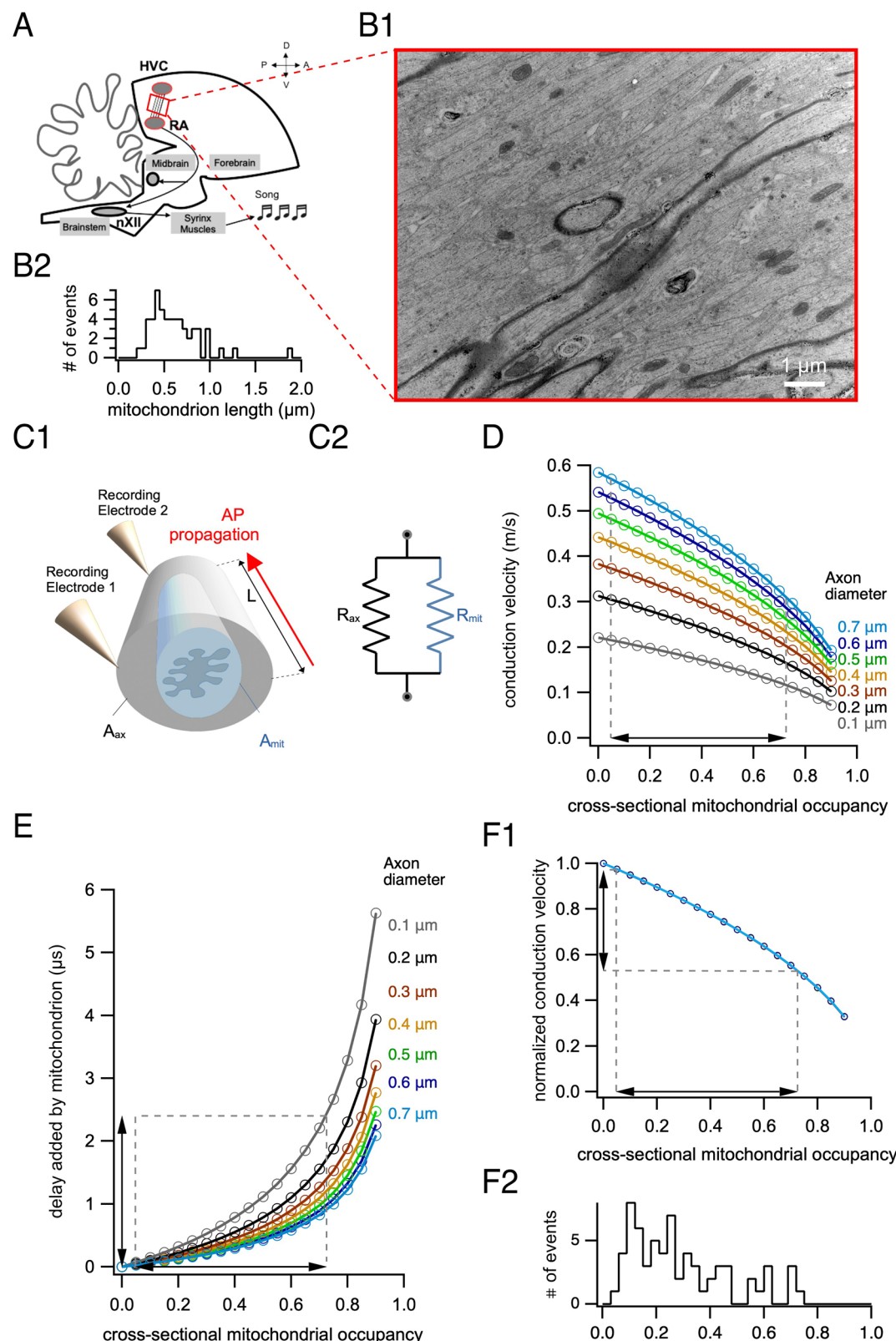

with 0.25 cross-sectional occupancy, and ~0.1 µs for larger 0.6 µm-diameter axons with 0.15 cross-sectional occupancy.

Although conduction velocity for a given cross-sectional mitochondrial occupancy was larger for larger diameter axons (Fig. 2D), the relative decrement in velocity due to cross-sectional mitochondrial occupancy was the same for all axon diameters examined (Fig. 2F1). That is, when

conduction velocity was normalized to that of the mitochondrion-free axon for each diameter, the relative decrease of conduction velocity was similar for all the diameters.

This observation is consistent with the prediction that the conduction velocity of an unmyelinated axon is proportional to its *(diameter)*[½1,10]. So, for two axons with the same underlying biophysical properties and cross-

**Fig. 2 | Impact of a single mitochondrion on action potential conduction velocity.** **A** Schematic of the pathway between HVC and RA. Modified from[50]. **B1** Longitudinal section of an axon bundle. **B2** Histogram of measured mitochondrion lengths estimated from the major axis of an ellipse fit ($n$ = 55 mitochondria). Bin size = 0.05 μm. **C1** Schematic of the modeled 0.6 μm long section of an axon containing a mitochondrion. $A_{mit}$, cross-sectional area taken up by a mitochondrion; $A_{ax}$, area free from mitochondrion; L, mitochondrion length. **C2** Equivalent circuit of the axial component used in our model. $R_{mit}$, resistivity of the mitochondrion; $R_{ax}$, intracellular resistivity of the axon without the mitochondria. **D** Action potential conduction velocity decreases for cylinders containing a mitochondrion, as a

function of cross-sectional mitochondrial occupancy of the axon for different axon diameters. The range of cross-section occupancies measured in our data is indicated with dashed lines. **E** Action potential latency increases for cylinders containing a mitochondrion, as a function of cross-sectional mitochondrial occupancy of the axon. The range of cross-sectional occupancies measured in our data is indicated with dashed lines. **F1** Conduction velocity normalized to a mitochondria-free axon overlapped for all diameters. **F2** Histogram of measured cross-sectional occupancies ($n$ = 66 mitochondria). Bin size = 0.03. The corresponding range is marked with dashed lines on the graph above.

sectional mitochondrial occupancy, $p$, we can express the velocity of axon 2 in terms of that of axon 1 as follows:

$$v_2(p) = c(p)v_1(p)(d_2/d_1)^{1/2} \qquad (7)$$

where $v_1$, $d_1$, and $v_2$, $d_2$ are the conduction velocity and diameter of axon 1 and axon 2, respectively, and $c(p)$ is a function of $p$ that depends on the underlying biophysical properties but is independent of diameter. In particular, *c(p) depends* on internal resistivity that in turn depends on $p$. This relationship holds for the mitochondrion-free axons with $p = 0$. So, for a given value of $p$, dividing the velocity $v_2(p)$ by that of the mitochondrion-free axon $v_2(0)$ gives:

$$v_2(p)/v_2(0) = c(p)v_1(p)/(c(0)v_1(0)) \qquad (8)$$

which doesn't depend on axon diameter. Thus, for all axon diameters examined, the decrease in normalized velocity with $p$ will lie on the same curve.

### Longitudinal coverage of axons by mitochondria

The overall impact of mitochondria on the conduction velocity along the pathway will depend on the fraction of the axon length that contains mitochondria. We estimated the average longitudinal coverage of the axon by mitochondria used in our simulations from electron micrographs with the following equation

$$\begin{aligned} \textit{total volumetric occupancy} &= \textit{cross-sectional occupancy} \\ &\times \textit{longitudinal coverage} \end{aligned} \qquad (9)$$

where *total volumetric occupancy* is the total fraction of the volume of unmyelinated axons in the micrograph occupied by mitochondria, *longitudinal coverage* is the fraction of axon length covered by mitochondria in the longitudinal section and *cross-sectional occupancy* is the fraction of axonal cross-section area covered by mitochondria as measured in Fig. 1. Hence, the total volumetric occupancy is proportional to the cross-sectional mitochondrial occupancy and the average longitudinal mitochondrial coverage.

We measured the fraction of the unmyelinated fibers occupied by mitochondria from a bundle sectioned longitudinally (185.50 μm² area × 0.06 μm thickness); the resulting total volumetric occupancy equalled 0.0377, that is, ~3.8%. Hence, the total occupancy in HVC$_{RA}$ axons fell within the range of previously measured average total occupancy of axons by mitochondria in mammals, e.g., ~2% in cerebellar parallel fibers and optic nerve axons, ~8% in olfactory receptor neurons, ~6% in the fornix, ~4% in retinal axons and; ~3 to 9% in different axons of hippocampal neurons[8,19,20]. Applying Eq. 9, *longitudinal coverage = total occupancy/cross-sectional occupancy* = 3.77/29.3 = 0.129 or ~13% on average.

### Mitochondrial increase in latency of action potential propagation between HVC and RA: dependence on mitochondrial longitudinal coverage and cross-sectional occupancy

We next investigated the impact of mitochondria on conduction velocity as action potentials propagate from HVC to RA. In order to examine the dependence of action potential conduction velocity in axons on both

cross-sectional occupancy and longitudinal coverage, we simulated a broad range of values. The longitudinal coverage of axons by mitochondria was varied in these simulations from 0 to 30%. This range includes the 13% longitudinal coverage measured above as well as that of axons measured in other systems, and the known changes in mitochondrial coverage with age, activity, injury, and regeneration[8,19–21]. The distance traveled by axons in a sagittal plane between HVC and RA was ~3 mm (Fig. 3A, B), similar to that estimated previously in another songbird, the zebra finch[22]. Note that the distance traveled by axons may be larger when linking distal parts of HVC and RA that are not in the same sagittal plane.

The presence of mitochondria relative to a mitochondria-free axon changed the amount of time required for an action potential to reach the RA for representative axonal sizes (Fig. 3D1–E3). For the typical 0.25 cross-sectional mitochondrial occupancy of a 0.4 μm diameter unmyelinated axon with 12.5% longitudinal coverage, action potentials accumulated an additional delay of 0.14 ms in their arrival to RA (Fig. 3E2) and a decrease in average conduction velocity of 2.1% (Fig. 3D2) from that of the mitochondria-free axon (0.442 m/s). For smaller fibers with 0.2 μm diameter, the typical 0.6 cross-sectional mitochondrial occupancy and 12.5% coverage, action potentials were delayed 0.85 ms (Fig. 3E1) and had a decrease in average conduction velocity of 8.1% (Fig. 3D1). For the 0.15 cross-sectional mitochondrial occupancy of a 0.6 μm diameter unmyelinated axon with 12.5% longitudinal coverage, action potentials were delayed by 0.06 ms in reaching RA (Fig. 3E3) and conduction velocity decreased by 1.2% (Fig. 3D3). Although the amount of additional delay depended on the axon diameter, the relative decrease in conduction velocity for a given cross-sectional occupancy and longitudinal coverage was similar for all diameters examined (Fig. 3D1-D3).

### Mitochondria induce inhomogeneous propagation of action potentials

The decrease in average conduction velocity along the pathway due to mitochondrial cross-sectional occupancy and longitudinal coverage is composed of small slowdowns as the action potential passes each individual mitochondrion. The amount of this local decrease depends on the properties of the mitochondrion. Hence, the conduction velocity along an axon with constant diameter and biophysical properties is not expected to be constant but to vary as mitochondria are encountered. Figure 4 shows an example of this inhomogeneity in action potential propagation in a simulated 0.3 μm diameter axon with 14% of the length containing mitochondria. The length of each mitochondrion was fixed at 1 μm, and the cross-sectional occupancy was drawn from a normal distribution with mean 0.3 and standard deviation 0.1 (Fig. 4A). This resulted in cross-sectional mitochondrial occupancies that fell within the observed range for a 0.3 μm diameter axon (see Fig. 1D2). The placement of the mitochondria along the axon was arbitrary. The figure shows the decrease in conduction velocity for each mitochondrion (Fig. 4B), but doesn't resolve the transition from mitochondrion-free to mitochondrion-containing axon sections.

### Discussion

We found that the canary motor pathway contains small-diameter unmyelinated axons that co-exist with myelinated axons in the axon bundles running between song nuclei HVC and RA, as previously shown in the

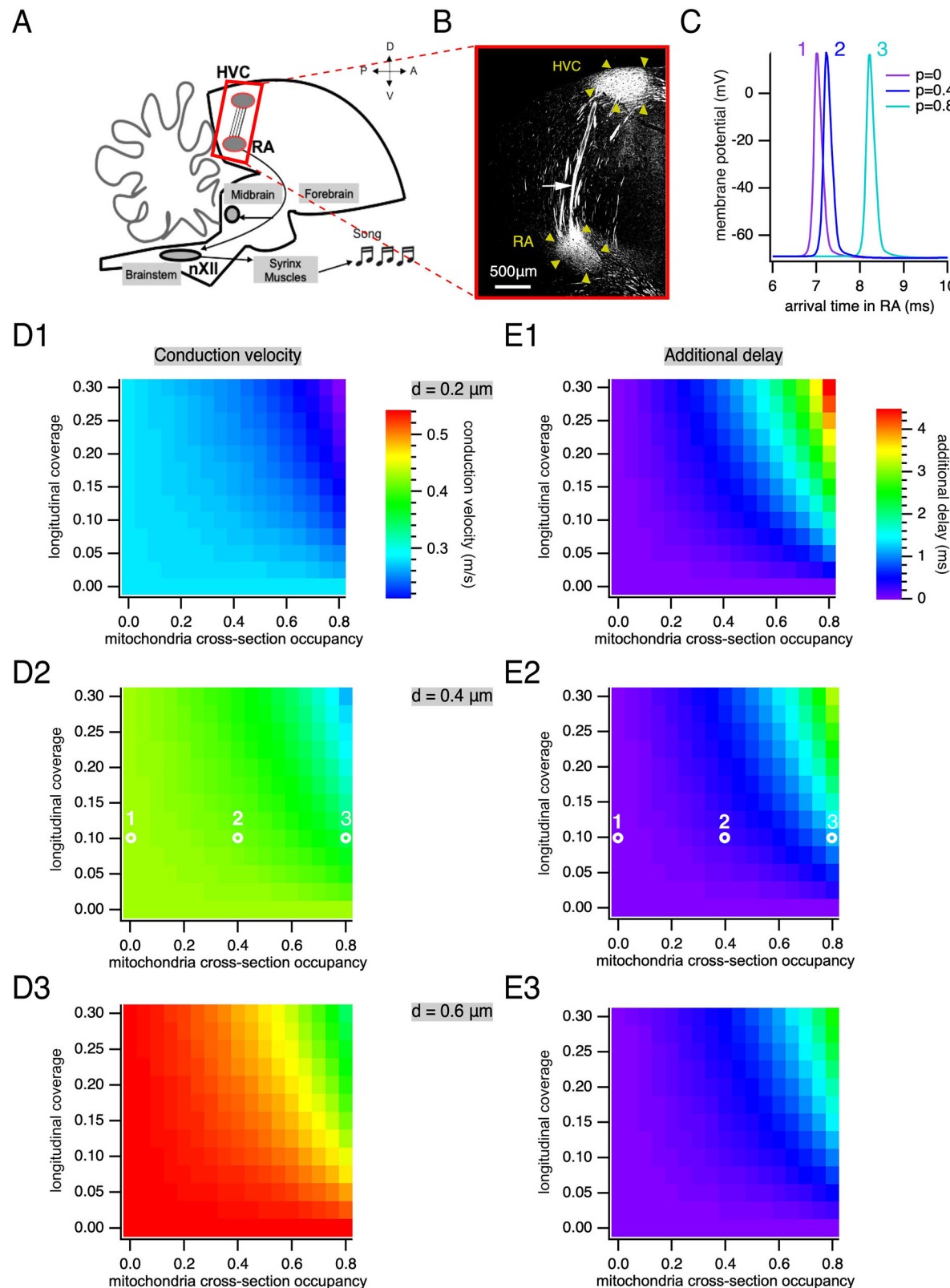

zebra finch[22]. We present measurements of the size and density of mito-chondria in unmyelinated axons from electron micrographs of these axon bundles from canaries. The mitochondria-containing unmyelinated axons ranged from ~0.2 to 0.7 μm diameter, and the cross-sectional occupancy of axons by mitochondria depended on the axon diameter: occupancy was larger for smaller diameter axons.

Our modeling showed that action potential propagation in axons is inhomogeneous due to the reduction of axoplasm by mitochondrial occu-pancy. Action potentials exhibit a local slowdown of propagation to about half of their original speed for the largest mitochondria-to-axon cross-sectional area ratios typically found in small axons. Although artificially induced inhomogeneities and changes in internal resistance have been

**Fig. 3 | Action potential propagation between HVC and RA and influence of mitochondria density. A** Schematic of the pathway along which we modeled the propagation of action potentials, between HVC and RA. Modified from[50]. **B** Immunostaining against neurofilament-associated antigen (3A10), which makes it possible to visualize the pathway formed by axons running in bundles between HVC and RA (indicated by yellow arrowheads). The white arrow indicates an axon bundle. **C** Simulated action potentials after propagating 3 mm along 0.4 μm diameter axons with 10% longitudinal coverage and different cross-sectional mitochondrial occupancies ($p$) ($p = 0$ in purple, $p = 0.4$ in dark blue and $p = 0.8$ in turquoise) showing increased time needed to reach the RA with increased mitochondrial occupancy (times measured from current stimulus). **D1** Action potential conduction velocity as a function of longitudinal coverage and cross-sectional mitochondrial occupancy for 0.2 μm diameter. **D2** Same as **D1** for 0.4 μm diameter axons. White circles indicate the parameters of action potentials shown in (**C**). **D3** Same as **D1** for 0.6 μm diameter. **E1** Additional delay of action potential arrival in RA as a function of longitudinal coverage and cross-sectional mitochondrial occupancy for 0.2 μm diameter axons. **E2** Same as E1 for 0.4 μm diameter. White circles indicate the parameters of action potentials shown in (**C**). **E3** Same as E1 for 0.6 μm.

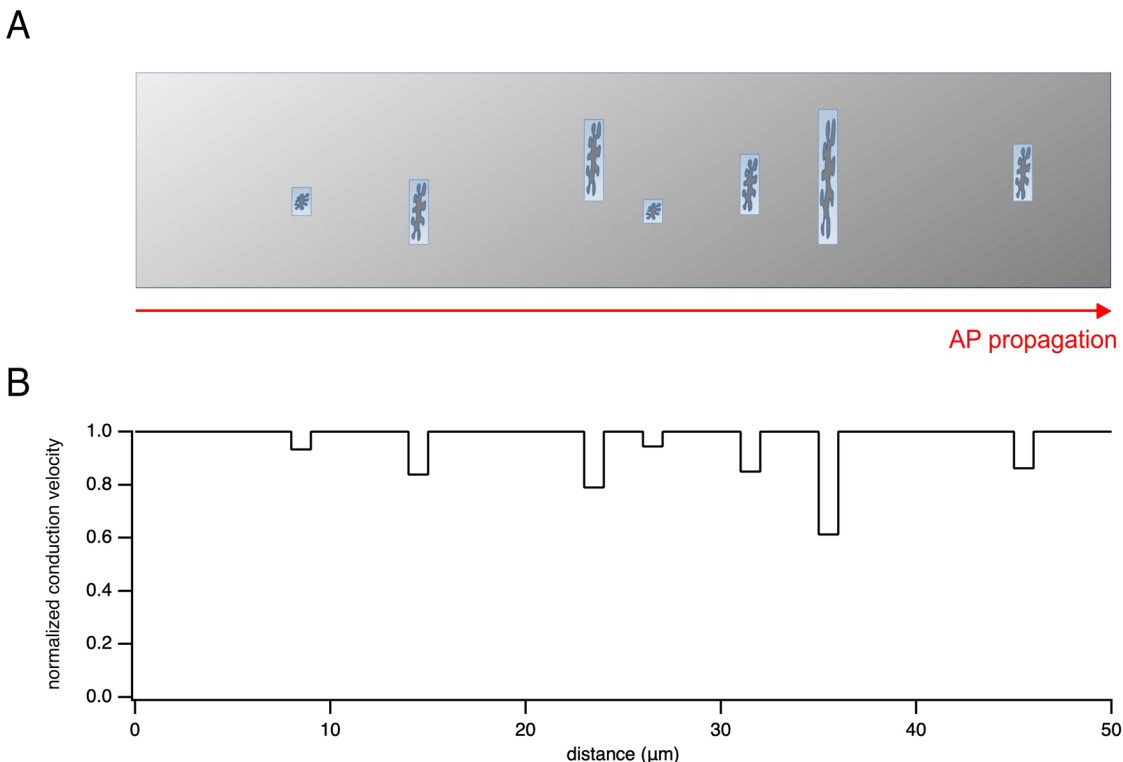

**Fig. 4 | Model of inhomogeneous propagation of action potentials along axons.** Schematic diagram illustrating the inhomogeneous propagation of action potentials along axons. **A** Axon where the axon diameter has been disproportionately enlarged to accurately illustrate the cross-section occupied by representative mitochondria. **B** Simulation of a 0.3 μm diameter axon with 14% longitudinal coverage by mitochondria. Due to the presence of mitochondria occupying part of the axonal volume, action potential propagation is locally slowed. The cross-sectional occupancies of the mitochondria were randomly distributed.

shown to induce changes in the charging speed of the plasma membrane[23,24], to the best of our knowledge, inhomogeneities of action potential propagation in unmyelinated fibers have not been reported so far. Indeed, whereas inhomogeneity of action potential propagation has been reported in myelinated fibers as 'saltatory' conduction, whereby conduction velocity increases at the internodes, action potentials have been assumed to propagate homogeneously along an unmyelinated axonal branch of constant diameter. However, we found that each time a propagating action potential encountered a mitochondrion, the instantaneous conduction velocity dropped by 16% of its original value for an average 0.29 cross-sectional occupancy, and by up to 47% for the largest mitochondrial cross-sectional occupancies that we observed (Fig. 2F1). This phenomenon does not affect all axons equally, given the modest scaling of mitochondrial size with axon size. Notably, small axons are more prone to a larger cross-sectional coverage, and thereby, to a larger local decrease in conduction velocity per mitochondrion. This modest scaling of the size of mitochondria with axon diameter may be constrained by their evolutionary origin and their co-evolution within eukaryotic cellular structures[25]. The minimum size of axonal mitochondria may therefore impose a lower limit on the smallest possible axonal diameter able to accommodate a mitochondrion, adding to previously described constraints on axon diameter[26].

The non-uniform propagation of action potentials due to mitochondria may lead to additional nonlinearities when combined with features such as axonal tree branching and impedance mismatches[27], ion channel clustering, or leakage through axonal gap junctions[5,28]. Note that, unlike changes that affect both the axial resistance and the plasma membrane, mitochondria locally change axial resistance without affecting plasma membrane capacitance or resistance. A simpler model that would only change the axonal cross-section[29] to take into account a mitochondrion with an infinite resistance would not keep the capacitance unchanged.

The small axons found in the songbird premotor pathway linking HVC and RA are known to conduct a neural code that relies on submillisecond precision[15]. Mitochondrial delays, which we estimated to be on the same time scale, are likely to constitute a limitation that the system has to take into consideration in its computations. Indeed, although submillisecond precision in the coordination of cell assemblies encoding song is necessary, a slight decrease in conduction velocity that delays the arrival of action potentials in RA within the millisecond range seems to pose a constraint that

the system can accommodate. Other features, such as miniaturization and volumetric constraints[8] may provide stronger selective pressure to keep small unmyelinated axons in this pathway. However, systems that rely strongly on submillisecond precision combined with fast conduction may be subject to stronger selective pressure, whereby mitochondria-induced slowdown could not be tolerated since it would significantly alter the neural code necessary for survival. An example of such a system that is fine-tuned for speed can be found in the auditory system[30], which is formed by large myelinated axons from cells with somata in the cochlear nucleus and axons that project to the main nucleus of the trapezoid body[31]. The slowdown by mitochondria can be mostly overcome by larger axons, since the fraction of the axonal cross-section occupied by mitochondria decreases as axon diameter increases, and by myelination[4,5,32]. This combination seems to be a solution that reduces mitochondria-induced action potential conduction delays.

Our results point toward an interesting paradox: on the one hand, energy, which as a first approximation can be estimated to be proportional to mitochondrial volume, is required to sustain axonal function, and in particular action potential generation and propagation[8,33]. Moreover, generating action potentials at higher frequencies leads to higher information encoding axons and likely requires larger mitochondrial volumes along axons[8]. Finally, unmyelinated axons may require high channel densities to overcome channel noise[34] and jitter[35], which will imply high energy demands to be fulfilled by axonal mitochondria. On the other hand, higher energy demands likely lead to higher mitochondrial density, and thereby, to slower conduction velocity for small-diameter unmyelinated axons. Thus, we propose that there is a tradeoff between energy requirements for information coding by axons and conduction velocity. Interestingly, changes in metabolic demand or efficiency could lead to plasticity in mitochondrial density and paradoxically to the slowing of action potentials as a cost for increased energy supply.

Our computational model only explored resistive changes in axial impedance, leaving aside contributions of mitochondrial capacitance. Our estimate of mitochondrial resistivity was consistent with the measurements of mitochondrial resistance of Jonas et al.[18] and is likely to be an underestimate because of our assumptions about the properties of the resistor (see section on model derivation in Results). However, increasing the mitochondrial resistivity would have only a minor effect on the results, since for a given occupancy ($p$), the equivalent resistivity ($R_{eq}$) of the combined axoplasm and mitochondrion saturates for large values of mitochondrial resistivity. The charging of mitochondrial capacitance seems to make a negligible contribution to the axial impedance change along the axon, since Padmarai et al.[16] found little change in measured mitochondrial impedance for signals up to 10 kHz, that is, within neuronal computational time-scales, justifying our assumption. Finally, we assumed a simple, cylindrical shape for a mitochondrion, which agrees with the low complexity index of mitochondria in hippocampal axons reported in Faitg et al.[20] The length and diameter of individual mitochondria found in axons, as well as their position, constitute heterogeneities that may locally modulate the magnitude of the action potential slowdown.

Does changing the length of a mitochondrion, while keeping the total longitudinal coverage constant, affect the average conduction velocity? In most of our simulations, the mitochondria had a fixed length of 1 µm and were distributed uniformly along the axon. This assumption had a minimal effect on the average conduction velocity for cross-sectional mitochondrial occupancies in the experimentally-observed range of $0.05 \leq p \leq 0.73$ (Fig. 2F2). In particular, for mitochondria lengths between 0.5 µm and 10 µm with 10% longitudinal mitochondrial coverage, the relative change in average conduction velocity from that of uniformly distributed 1 µm long mitochondria was less than 0.6% for $p \leq 0.75$. Similarly, if the 1 µm long mitochondria were distributed stochastically, which permits some variability in mitochondria length along the axon, the relative change in conduction velocity was <0.2%.

Finally, it should be considered that mitochondria play a role in maintaining calcium homeostasis. Jonas et al.[18] reported large conductance changes in mitochondria in the presynaptic terminal with ongoing stimulation. Thus, the resistivity of mitochondria at the terminal is a dynamic parameter, and the conductance state of mitochondria in the axon may strongly depend on calcium increases. However, large changes in mitochondrial permeability due to cytosolic calcium increases are unlikely to occur along the axon, which expresses a lower density of calcium channels. Thus, while changes in mitochondrial conductance are unlikely to introduce major changes in our results[36], they can easily be implemented in our two-conductance-pathways model, including the likely large increases in conductance related to neurotransmitter release events at the terminal.

How widespread is the phenomenon reported here in axons across phylogeny? First, long, small-diameter axons are widely found in mammalian central nervous systems[9,12,13], including humans[14]. Second, axonal occupancy by mitochondria is in the same range as reported here in both invertebrates and vertebrates[9,37], suggesting that the phenomenon is widespread across the phylogeny. If we consider the ~0.1–0.8 ms delays in action potential propagation expected for 3 mm long axons, and how these delays may generalize to axons one to two orders of magnitude longer, we would expect cumulated latencies in the millisecond to tens of milliseconds range, e.g., in the long unmyelinated fibers found in the human brain[14]. Remarkably, mitochondria show a large degree of plasticity with activity, age, or disease[20,21]. An interesting application of our model would be to compute the magnitude of changes in action potential conduction velocity induced by previously-reported mitochondrial plasticity, e.g., during aging and in disease[20]. These should be of particular relevance to the fields of axonal function, degeneration, and regeneration.

The presence of additional organelles, not only of mitochondria, may similarly contribute to slowing action potential conduction and increase the overall impact on conduction velocity. Additionally, it would be interesting to model the impact of mitochondria in dendrites, which show a larger total mitochondrial occupancy than axons, e.g., in neocortical pyramidal cells[38]. There, mitochondrial occupancy may also interact with the propagation of synaptic events and dendritic spikes[39]. Finally, we have focused on modeling the impact of mitochondria in single axons. It remains to be established how delays induced by mitochondria are affected by inter-axonal interactions in bundles. Indeed, the presence of electrical interactions between axons, such as those due to ephaptic or gap junction-mediated electrical transmission may influence the activity of nearby axons[5,40]. One might hypothesize that if axons were excited simultaneously, these mechanisms could provide additional excitation that might counteract the slowdown due to mitochondria, as long as the mitochondria in neighboring axons did not occur in the same longitudinal position along the bundle. The scenario of nearby cells firing simultaneously in the premotor songbird axon bundles seems unlikely, since $HVC_{RA}$ cells have been described as extremely sparse-firing, covering a wide range of temporal windows of ~10 ms each within songs lasting seconds[15,41,42]. However, this effect may be particularly interesting to explore in networks with large levels of synchrony, where the relative distribution of mitochondria in nearby axons may play a major role in the magnitude of these interactions.

So far, cable theory has focused on biological cables free of organelles[11]. Including intracellular organelles will add to our understanding of electrical signal propagation along biological cables found in neurons as well as in other cell types. Introducing mitochondria and more generally, intracellular organelles, as part of the structural design of axons could improve the accuracy of models for action potential propagation, which could prove crucial to understanding specialized systems containing small-diameter axons.

## Methods
### Animals
Three adult male canaries (*Serinus canaria*, > 240 days post-hatch) housed in outdoor aviaries at the Max Planck Institute for Ornithology (Seewiesen) were euthanized by an overdose of isoflurane (two for electron microscopy and one for light microscopy). We have complied with all relevant ethical regulations for animal use. Housing, welfare of the animals, and

experimental procedures complied with the requirements of the European Directive on the protection of animals used for scientific purposes 2010/63/EU of the European Parliament and the German Animal Protection Act.

## Fixation

For electron microscopy, after death had been confirmed, animals underwent intracardiac perfusion for 2 min with a phosphate buffer saline (PBS) solution containing sodium nitroprusside (VWR chemicals, 10 µg/ml) followed by 20 min with a "Karlsson-Schultz" perfusion solution containing 4% formaldehyde (Carl Roth Art. Nr. 0335), 2.5% glutaraldehyde (Electron Microscopy Sciences, cat.# E16220), 0.5% NaCl in phosphate buffer adjusted to pH 7.4[43]. Perfusion speed was 1 ml/min. Brains were post-fixed for 24 h at 4 °C.

For light microscopy, after death had been confirmed, the canary underwent intracardiac perfusion for 2 min with a PBS solution containing sodium nitroprusside (VWR chemicals, 10 µg/ml), followed by 20 min with a perfusion solution containing 4% formaldehyde (Carl Roth Art.Nr. 0335) in PBS. Perfusion speed was 1 ml/min. The brain was postfixed for 24 h at 4 °C.

## Electron microscopy

Sagittal sections 100 µm to 300 µm thick were sliced with a vibratome (Leica VT1200S). Sections approximately 1 mm by 1 mm isolated from the region containing bundles that exit the nucleus HVC in the direction of RA were further processed for electron microscopy. Osmification was performed with 1% Osmium Tetroxide (Electron Microscopy Sciences, cat.#19152) in 0.1 M Sodium Cacodylate pH 7.4 (Electron Microscopy Sciences, cat.#11653), for 40 min. Osmification was followed by 3 rounds of washing in distilled water and dehydration in successive steps, each of 10 min, in 30%, 50%, 70%, and 100% Ethanol. Samples were embedded in Spurr's low viscosity embedding kit (Electron Microscopy Sciences, cat.#14300) according to the manual (https://www.emsdiasum.com/docs/technical/datasheet/14300) for 48 h at 60 °C. Semi-thin sections were cut at 0.5 µm thickness with an ultramicrotome EM UC6 (Leica) and stained with epoxy tissue stain (Electron Microscopy Sciences, cat.#14950). Ultra-thin sections were cut at 60 nm thickness with an ultramicrotome EM UC6 (Leica). Sections were stained with 'ultrostainer' (Leica) with 0.5% uranyl acetate (Electron Microscopy Sciences, cat.#22400-1) and 3% lead citrate (Ultrostain 2, Leica).

Images were acquired with a JEOL (JEM-1230) transmission electron microscope and a Gatan SC1000 Orius CCD Camera with the software Gatan DigitalMicrograph™ version 2.3. Image quantifications were performed on magnifications of 40000 to 80000 from axon bundles between HVC and RA. Images were analyzed with ImageJ software (https://imagej.net/software/fiji/) and Igor Pro software (https://www.wavemetrics.com/). We fitted the axons and mitochondria to ellipses with the ImageJ built-in plugin. Occupancy of axonal cross-sections by mitochondria was measured as the ratio of the area estimated from a disk whose diameter was set to the minor axis of an ellipse fitted to the mitochondrion to that analogously estimated for the corresponding axon. In three axon cross sections, we found two to three mitochondria. These were excluded from the mitochondria-to-axon ratio calculations based on the ellipse fits and the correlation of axon and mitochondria diameters.

The total volumetric occupancy was estimated from the measurement of the volume occupied by unmyelinated axons inside a bundle in longitudinal sections of bundles. Large fractions of extracellular space found in the bundle were deducted from this volume.

## Light microscopy

Brain sagittal sections 60 µm thick were sliced with a vibratome (Leica VT1200S) *and stained as follows:* a sagittal slice containing nuclei HVC and RA was incubated in a blocking solution (BS) containing 1% bovine serum albumin (weight/volume), 0.1% saponin and 1% triton X-100 in PBS at room temperature for 1 h. The slice was then incubated with the primary antibody against the neurofilament-associated antigen 3A10

(Developmental Studies Hybridoma Bank, mouse IgG1, dilution 1:130 in BS) at 4 °C for 48 h, washed four times in BS at room temperature, and incubated in DyLight anti-mouse IgG1 (Dianova, 115-475-205, dilution 1:100 in BS) at 4 °C for 24 h, washed once in BS and three times in PBS and mounted in Vectashield mounting medium.

Image acquisitions were performed at the Center for Advanced Light Microscopy (LMU) with a Nikon Ti-E microscope equipped with a CSU-W1 spinning disk confocal unit, an Andor Borealis illumination unit, an Andor ALC600 laser beam combiner, a CFI Apochromat LWD Lambda S 40XC WI objective, and an Andor iXon Ultra 888 EMCCD camera. The fluorochrome was visualized with an excitation wavelength of 405 nm (emission filter 419–465 nm). Images were acquired with a pixel size of 326 nm. Acquisitions were performed with a NIS-Elements software (version 5.02.00).

## Computational modeling

Models for the unmyelinated axons of canary $HVC_{RA}$ cells were simulated using the NEURON simulation environment (version 7.4)[44,45]. Based on the morphology of these axons, the simple ball and stick model neuron consisted of a spherical soma 6 µm in diameter with a single cylindrical axon less than 1 µm in diameter. Simulated axon diameters ranged from 0.1 µm to 0.7 µm, in agreement with experimental measurements reported here. The soma contained only a passive leak conductance, while the axons contained fast sodium and delayed rectifier potassium conductances in addition to the leak conductance. The descriptions of the ion channel kinetics were taken from the model for mammalian neocortical pyramidal axons of Cohen et al.[33] available from the ModelDB database (https://modeldb.science/260967). The fast sodium channel has the 8-state kinetic gating scheme of Schmidt-Hieber and Bischofberger[46]. The potassium channel kinetics were described using the Hodgkin-Huxley formalism[10] for a non-inactivating potassium channel with parameters based on a Kv1.1 subunit[47]. The ion channel densities were set to 1000 pS/cm$^2$ for the sodium channel and 3000 pS/cm$^2$ for the potassium channel, and were uniform along the axon. These channel densities were chosen to fit the amplitude and shape of the spike waveform of $HVC_{RA}$ cells[48] and the range of conduction delays measured from HVC to the RA along axons putatively identified as unmyelinated[22,49]. The simulations were run at 40 °C according to physiological canary body temperatures. The cytoplasmic resistivity of the axon without mitochondria was set to 100 Ωcm, while that of the sections containing a mitochondrion was varied to model the effect of the mitochondrion (see Results). For all sections, the specific membrane capacitance was set to 1 µF/cm$^2$, and the reversal potential of the leak current to −70 mV.

The axon was made up of two types of sections: those containing a mitochondrion and those with just axoplasm. Generally, the mitochondrion-containing sections were 1 µm in length and were distributed uniformly along the axon. The effect of varying the length and distribution of the mitochondrial sections on the average computed action potential conduction velocity was usually small as long as the total amount of mitochondria in the axon remained fixed. In particular, when the 1 µm mitochondrion sections were distributed randomly instead of uniformly, the relative difference in the conduction velocity was less than 0.2%. However, in the extreme case when very large mitochondria were collected in one long section, the relative change in conduction velocity was ~11%. Conduction velocity was computed from the difference between the times when the action potential upswing crossed −5 mV at two positions near the middle of the axon separated by a known distance, typically 200 µm. The time when the membrane potential crossed −5 mV was interpolated from the times of adjacent points on the voltage trajectory spanning −5 mV. To compute the local effect on conduction velocity of a single mitochondrion, the simulation was run with a single long mitochondrion containing axon section to minimize numerical errors due to the boundary conditions. The conduction velocity of an axon without mitochondria was similarly computed with a single long axon section. The time required for an action potential to propagate a given distance (e.g., 3 mm from HVC to RA) was estimated from the computed conduction velocity. The difference between

the times necessary to travel a certain distance along an axon containing mitochondria and one without mitochondria was reported as the additional delay due to mitochondria.

The simulations used NEURON's default backward Euler integration with a time step size of 2.5 μs. The soma was subdivided into 2 μm compartments; the spatial grid for the axon was finer, typically 0.33 μm and 0.82 μm compartments for axon sections with and without mitochondria, respectively. A 3-fold increase in the fineness of the spatial grid resulted in a relative change in the conduction velocity of less than 0.01%. Action potentials were elicited by a 0.5 ms duration 0.5 nA current pulse to the soma following a 10 ms delay to allow any membrane voltage transients to settle back to the resting membrane potential.

### Statistics and reproducibility
Statistical tests were performed in Igor Pro software (version 8.0) and R (version 4.4.1). All tests were two-sided. Data is reported as mean ± standard deviation. Curve fits and confidence intervals for Fig. 1 were performed with Igor Pro.

### Reporting summary
Further information on research design is available in the Nature Portfolio Reporting Summary linked to this article.

### Data availability
Original data is available from the corresponding author upon reasonable request. The source data behind the graphs in the paper can be found in Supplementary Data 1.

### Code availability
The NEURON model for mitochondria containing axons used to generate the data for Fig. 3 will be available from ModelDB upon publication (https://modeldb.science/2019788).

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

## Acknowledgements
We thank Manfred Gahr for support, feedback, and for providing excellent infrastructure throughout the study and Benedikt Grothe for providing excellent infrastructure. We are thankful to Marianne Braun for processing the samples for electron microscopy and for advice throughout the course of this study, and the animal caretakers at the Max Planck Institute for Ornithology for excellent care of the canaries; Wiebke Möbius and Moritz Hertel for advice on perfusions. The authors thank the Center for Advanced Light Microscopy at the Ludwig-Maximilians-Universität München for support. We thank Romain Brette and Angelika Harbauer for their feedback on previous versions of this manuscript. The 3A10 antibody was obtained from the Developmental Studies Hybridoma Bank. This work was supported by the Max Planck Society, the LMU Munich and in part by a grant from the Cades Foundation, Honolulu, Hawaii (A.C.). This is the University of Hawai'i at Mānoa School of Ocean and Earth Science and Technology contribution number 11982.

## Author contributions
A.C. conceptualized the model, performed simulations, analyzed simulations, wrote the manuscript. P.A. conceptualized the experiments and the model, performed experiments, analyzed experimental data and simulations, wrote the manuscript.

## Funding

## Competing interests
The authors declare no competing interests.
