## [Transparent Peer Review file · Communications Biology]

Mitochondria delay action potential propagation

Corresponding Author: Dr Pepe Alcami

Version 0:

Reviewer comments:

Reviewer #2

(Remarks to the Author)

This manuscript combines anatomical work on mitochondria in thin, unmyelinated axons with careful computational modeling of axonal conduction velocity for action potentials. And this in a system in which high temporal precision is a requirement. Hence, the question addressed is interesting.

My recommendation would be for the authors to compare and contrast to an even simpler reductionist description along the classical lines of AC Scott Rev. Mod. Phys. 47, 487 (1975). Treat the mitochondrion as being of infinite resistance, hence the change is only and solely due to a change in the cross-section. The scaling laws predict that

$$v \sim A^{1/4}/C$$

where A is the cross-section, C the capacitance. The current treatment of the subject in the manuscript seems correct, but somewhat dense. I think the argument can be made even more cogent.

Moreover, I think a crucial question is whether the variations in the speed of action potentials due to the "road bumps" created by mitochondria might not be mitigated by ephaptic coupling across nerve fibers. This would presume that the mitochondria are not distributed at exactly the same locations in different axons.

Could the authors address this question?

Reviewer #3

(Remarks to the Author)

This manuscript examines the delay in action potential propagation along the axon that is introduced by the presence of swellings due to mitochondria in unmyelinated axons from birds. In a first section, they show using electron microscopy that mitochondria occupy about 30% of the axon in cross sections. Then, they show using computational modeling that the action potential conduction velocity is reduced by 16-40% when a mitochondria is encountered.

This is a nice and solid study, well written and designed. I have not found any serious weakness.

Version 1:

Reviewer comments:

Reviewer #3

(Remarks to the Author)

The authors have well addressed the points raised in the first round of evaluation. I have no further comment.

RESPONSES TO THE REVIEWERS COMMENTS

Please find our point-by-point response to the reviewers' comments below. The reviewers' comments are given below in black text and our responses are indicated by blue text. We have included our changes to the manuscript with the corresponding line numbers (blue text).

We would like to thank the reviewers for their careful reading of our manuscript and their helpful comments.

In addition, we made some minor improvements in the writing, corrections of typos and methods, and adjusted the quantifications to the checklist requirements from Communications Biology. All changes have been indicated in the manuscript using Track Changes.

Reviewers' comments:

Reviewer #2 (Remarks to the Author):

I.1. This manuscript combines anatomical work on mitochondria in thin, unmyelinated axons with careful computational modeling of axonal conduction velocity for action potentials. And this in a system in which high temporal precision is a requirement. Hence, the question addressed is interesting.

We would like to thank Reviewer 2 for the positive comments.

I.2. My recommendation would be for the authors to compare and contrast to an even simpler reductionist description along the classical lines of AC Scott Rev. Mod. Phys. 47, 487 (1975). Treat the mitochondrion as being of infinite resistance, hence the change is only and solely due to a change in the cross-section. The scaling laws predict that

$$v \sim A^{1/4}/C$$

where A is the cross-section, C the capacitance. The current treatment of the subject in the manuscript seems correct, but somewhat dense. I think the argument can be made even more cogent.

We appreciate the suggestion and we included a section (lines 426-428) in the discussion. We would like to justify our modeling approach for the following reasons:

1. There is a major difference between the model we proposed in which the presence of a mitochondrion modifies the intracellular resistivity of the axon but doesn't change its cross-section and one in which the effect of a mitochondrion is to reduce the axon's cross-sectional area. The reduction in axonal cross-section will affect the membrane surface area and hence, the membrane capacitance. Our model assumes that the mitochondrial pathway does not affect the axonal plasma membrane, and thereby, does not affect the membrane capacitance.
2. Our aim was to develop a model where mitochondrial conductance is variable and realistic. Indeed, although axonal mitochondria are typically very resistive, they can also increase their conductance significantly, particularly in association with vesical release events and calcium

buffering (e.g. in axon terminals (Jonas et al. Science 286, 1347 (1999)). Our model can be easily modified to include varying conductance values of mitochondria, whose role has recently been reevaluated to include not only providing energy, but also to acting as major ionic buffers, as well as some being metabolically inactive (Hirabayashi et al. BioRxiv 2024.02.12.579972 (2024)).

3. In order to determine whether the decrease in conduction velocity due to mitochondria was enough to produce delays on the time scale of the songbird system, we needed an estimate for the mitochondrial resistivity that we could justify. Although the resulting delays were similar to those that result from an infinite mitochondrial resistivity (i.e. the limit as mitochondrial resistivity goes to infinity), they were produced using a conservative estimate for the resistivity, which supports the biological significance of the phenomenon.

In order to consider the reviewer's comment, and to clarify the justification of our two-conductance pathways model, we have added the following text to the Results and Discussion sections:

Lines 149-152: 'To model the impact of a mitochondrion on conduction velocity, we made the simplifying assumption that the mitochondrion acts like a region of high resistance to axial current flow leaving unchanged the axolemma resistance and the axonal cross-section, and thereby, the axonal membrane capacitance.'

Lines 179-183: 'In particular, if the resistivity of the mitochondrion is sufficiently large that the current path through the mitochondrion can be ignored, then $R_{eq} = R_{ax}/(1-p)$, the limit of R_{eq} as R_{mit} goes to infinity. So for a given value of $p < 1$, R_{eq} saturates as R_{mit} gets large. On the other hand, if the mitochondrion fills the axonal cross-section, then $p = 1$ and $R_{eq} = R_{mit}$.'

Lines 424-428: 'Note that unlike changes that affect both the axial resistance and the plasma membrane, mitochondria locally change axial resistance without affecting plasma membrane capacitance or resistance. A simpler model that would only change the axonal cross-section [30] to take into account a mitochondrion with an infinite resistance would not keep the capacitance unchanged.'

Lines 498-501: 'Thus, while changes in mitochondrial conductance are unlikely to introduce major changes in our results [37], they can easily be implemented in our two-conductance-pathways model, including the likely large increases in conductance related to neurotransmitter release events at the terminal.'

I.3. Moreover, I think a crucial question is whether the variations in the speed of action potentials due to the "road bumps" created by mitochondria might not be mitigated by ephaptic coupling across nerve fibers. This would presume that the mitochondria are not distributed at exactly the same locations in different axons.

Could the authors address this question?

We thank the reviewer for raising this point. Our article examines the question of how mitochondria affect conduction velocity at the single axon level, and not at the network level. However, the reviewer is right to call attention to the network level and how it may influence single axons given that a feature of the axons characterized in this article (linking song nuclei HVC and RA) is their spatial arrangement in bundles. Axon bundles, by bringing axons in close vicinity, have indeed been suggested to generate

ephaptic interactions between axons. We have included a section (lines 523-536) in the discussion to address possible larger scale network mechanisms taking into account inter-axonal interactions.

The very sparse firing of HVC cells projecting to RA, with few cells synchronous across the whole population, renders unlikely that two nearby axons are firing in synchrony, which may intuitively be a way by which ephaptic currents in nearby axons may compensate for the action potential delay induced by mitochondria. The question may be relevant in other networks, and the outcome of how ephaptically-induced delays or accelerations interact with mitochondrial delays, and under which circumstances they may significantly counteract mitochondrial delays is fascinating, but falls beyond the single-axon characterization of this article and risks further complicating a phenomenon that, in single axons, and in our network, seems to us unlikely to be affected by ephapses.

Lines 517-536: 'The presence of additional organelles, **not only of mitochondria**, may similarly contribute to slowing action potential conduction and increase the overall impact on conduction velocity **discussed here**. **Additionally**, it would be interesting to model the impact of mitochondria in dendrites, which show a larger total mitochondrial occupancy than axons, e.g. in neocortical pyramidal cells [39]. There, mitochondrial occupancy may also interact with the propagation of synaptic events and dendritic spikes [40]. **Finally, in this article we have focused on modeling the impact of mitochondria in single axons**. It remains to be established how delays induced by mitochondria are affected by inter-axonal interactions in bundles. Indeed, the presence of electrical interactions between axons, such as those due to ephaptic or gap junction-mediated electrical transmission may influence the activity of nearby axons [5, 41]. One might hypothesize that if axons were excited simultaneously, these mechanisms could provide additional excitation that might counteract the slowdown due to mitochondria, as long as the mitochondria in neighboring axons did not occur in the same longitudinal position along the bundle. The scenario of nearby cells firing simultaneously in the premotor songbird axon bundles seems unlikely, since HVC_{RA} cells have been described as extremely sparse-firing, covering a wide range of temporal windows of ~10 ms each within songs lasting seconds [15, 42, 43]. However, this effect may be particularly interesting to explore in networks with large levels of synchrony, where the relative distribution of mitochondria in nearby axons may play a major role in the magnitude of these interactions.'

Reviewer #3 (Remarks to the Author):

II. This manuscript examines the delay in action potential propagation along the axon that is introduced by the presence of swellings due to mitochondria in unmyelinated axons from birds. In a first section, they show using electron microscopy that mitochondria occupy about 30% of the axon in cross sections. Then, they show using computational modeling that the action potential conduction velocity is reduced by 16-40% when a mitochondria is encountered.

This is a nice and solid study, well written and designed. I have not found any serious weakness.

We would like to thank Reviewer 3 for the positive review.